# OpenReview forum: "Understanding Adversarial Transfer: Why Representation-Space Attacks Fail Where Data-Space Attacks Succeed"
_ICLR.cc/2026/Conference — Submitted to ICLR 2026_

### Official Review · Reviewer_J7K8 · 2025-10-27

**Soundness:** 1
**Presentation:** 2
**Contribution:** 1
**Rating:** 2
**Confidence:** 3

**Summary:**

This paper studies the difference between the transferability of attacks in "activation-space" and "data-space". The authors show that attacks transfer for the latter but not for the former across several settings and provide a simple theoretical justification for this. The authors also attempt to connect these insights to the observations that jailbreaks do not transfer across vision language models (VLMs).

**Strengths:**

- The paper targets the very interesting problem of explaining why jailbreaks do not transfer for vision language models.
- The distinction between "representation-space" and "data-space" attacks and their differing rates of transferability are clearly explained an thoroughly established across different settings and some theoretical justification is provided.

**Weaknesses:**

1) The paper is motivated by the observation that jailbreaks do not transfer in VLMs, however, the papers cited showed this in "data-space" attacks. Therefore, this paper hinges on the claim that image perturbations in VLMs are akin to "representation-space" attacks. This is an interesting claim which is unfortunately glossed over, as in the following excerpt:
>Implicit in our thinking is that text is the data for vision-language models, and from their “perspective”, visual inputs are effectively perturbations to their activations, akin to a Neuralink implant in a human brain.

2) If the connection to the lack of transferability of jailbreaks in VLMs has not been established as discussed in 1), the study of "representation-space" attacks becomes less significant, particularly because very limited context or motivation is provided for this kind of attack. Is this a kind of attack that has previously been studied in the literature? Why is this kind of white box vulnerability important?

**Questions:**

If the jailbreaks which do not transfer in VLMs are perturbations applied to the images, why are these considered akin to "representation-space" attacks and not "data-space" attacks? Crucially, even if two VLMs learn rotated versions of each others representations, I would still expect their representations of "dog" to depend on similar parts of the input so that it is not clear that a perturbation for one model will not also affect the other.

Can you provide more context and motivation for "representation-based" attacks, how do they tie to existing literature and why is their lack of transferability important?

---

> ### Author Response · Authors · 2025-11-21
> **Response to Reviewer J7K8**
>
> We thank Reviewer J7K8 for their rigorous questioning of our core premise. You ask the crucial question: *If the perturbation is applied to pixels (input), why do we classify it as a representation-space attack?*
>
> This distinction is not merely semantic; it is structural. We argue that the failure of transfer in VLMs is driven by the **misalignment of the interface** between the vision encoder and the language model.
>
> # Responding to Weakness 1 & Question 1: Why VLM Image Inputs are Representation Attacks
>
> To explain why we classify VLM image attacks as representation-space attacks, consider the **Lock and Key** analogy:
>
> 1.  **Text Attacks (Data Space):**
>     *   To jailbreak an LLM via text, the adversary must find a sequence of tokens (the "key") that forces the model to output harmful content.
>     *   Because LLMs share similar training data and tokenizers (or map to semantically similar clusters), the two have similar input-output maps and the "key" that unlocks Model A (e.g., the string "Sure, here is how...") also fits the lock of Model B. The shared "Data Space" ensures the keys are transferable.
>
> 2.  **VLM Image Attacks (Representation Space):**
>     *   In an adapter-based VLM, the "key" that enters the LLM is **not** the pixels. It is the sequence of continuous embedding vectors produced by the Projector.
>     *   We empirically show (Fig. 17) that due to random initialization and lack of constraints, the projection layers of different VLMs are **geometrically unaligned**.
>     *   Consequently, the embedding sequence "key" required to unlock Model A ($v_A$) has a completely different geometric shape than the "key" required for Model B ($v_B$).
>     *   An adversarial image perturbation is optimized to morph the image into $v_A$. When this perturbed image is fed into Model B, it produces a representation vector that is an unrelated, meaningless quantity. Since $v_A$ does not fit the lock of Model B, the attack fails.
>
> # Responding to the "Dog" Counterexample
>
> The reviewer correctly notes that semantic features (like "dog") rely on similar input parts. While both vision encoders may recognize a dog, the specific adversarial perturbation required to twist that "dog" embedding into a "jailbreak" vector is entirely dependent on the specific function of the projector.
>
> # Responding to Weakness 2 & Question 2: Motivation for Representation-Space Attacks
>
> The reviewer asks why we study representation-space attacks if they don't naturally occur in black-box settings. We study them for three reasons:
>
> 1.  **Explaining the “Paradox”:** The field has been puzzled by why adversarial examples transfer so well for ImageNet and LLMs but fail for VLMs (Schaeffer et al., 2024, Rando et al., 2024). Our framework resolves this “paradox”: VLM image attacks fail because they are functionally equivalent to soft-prompt attacks (which also don't transfer, as we show in Sec 4), due to the unaligned projector interface.
>
> 2.  **Scientific Control:** To prove that "alignment" is the variable controlling transfer, we needed to construct a setting where we could manually control alignment. Representation-space attacks allow us to do this. In Section 6, we show that when we *do* align the representations (via finetuning), these attacks suddenly start transferring.
>
> 3.  **Future Threat Modeling:** As modular AI systems become more common, understanding attacks on the internal interfaces (representations) becomes critical. Our work suggests that modular systems with unaligned interfaces possess a natural defense against transfer attacks.
>
> # Conclusion
>
> We admit that the initial draft "glossed over" this structural argument. We will rewrite the Introduction and Section 5 to explicitly use this "Shared Interface vs. Unaligned Interface" logic (and the Lock/Key analogy) to justify the classification. We believe this resolves the conceptual gap the reviewer identified.

---

### Official Review · Reviewer_DviV · 2025-10-30

**Soundness:** 3
**Presentation:** 3
**Contribution:** 3
**Rating:** 2
**Confidence:** 3

**Summary:**

This paper addresses a paradox: adversarial examples transfer between image classifiers/LLMs, but image jailbreaks do not transfer between VLMs.

**Strengths:**

- Well written.
- The proposed method is interesting.

**Weaknesses:**

- The proof uses $\ell_2$-optimal representation attacks, but the empirical experiments use $\ell_\infty$ constraints.
- VLM data-space attacks (text jailbreak) are tested without image inputs, but VLMs are multimodal.
- This paper claims "images behave like representation-space attacks " for VLMs, but provides no analysis of why this is the case.

**Questions:**

- Please see "Weaknesses".

---

> ### Author Response · Authors · 2025-11-21
> **Response to Reviewer DviV**
>
> We thank Reviewer DviV for their feedback. We understand the concern regarding the justification for treating images as representation-space attacks and the experimental choices regarding norms and text-only baselines. We address these points below.
>
> # Responding to Weakness 3: Why Images Behave Like Representation-Space Attacks
>
> The reviewer notes that we claim "images behave like representation-space attacks" but asks for an analysis of *why*. This is the central structural argument of our work, and we will expand Section 5 to make this explicit.
>
> **The Structural Reason:**
> In adapter-based VLMs (like the LLaVA and Prismatic models we study), the architecture consists of a Vision Encoder, a Projector, and an LLM Backbone.
>
> 1.  **Text Pathway:** Text input is tokenized into discrete integers from a shared vocabulary (e.g., the Llama tokenizer). These integers map to static lookup embeddings. This is the data space.
>
> 2.  **Image Pathway:** An image $x$ is processed by the encoder and projector to produce a sequence of continuous embedding vectors $v = P(E(x))$. These vectors are injected directly into the LLM, bypassing the discrete token lookup.
>
> From the perspective of the LLM (which performs the computation), the visual input is not "data" in the sense of the language it was trained on. It is a sequence of high-dimensional continuous vectors specific to the weights of the encoder $E$ and projector $P$. Because different VLMs use different random initializations for their projectors (and often different vision encoders), the resulting vectors $v_{model1}$ and $v_{model2}$ for the same image are **geometrically misaligned**.
> Therefore, an adversarial perturbation on the image acts effectively as a perturbation on these internal representations. Since the coordinate systems of $v_{model1}$ and $v_{model2}$ are unaligned (as we show in Figure 17), the attack fails to transfer, exactly matching the behavior of representation-space attacks (soft prompts) we analyze in Section 4.
>
> # Responding to Weakness 1: Mismatch between Theoretical $\ell_2$ and Empirical $\ell_\infty$ Norms
>
> The reviewer is correct that our theoretical proofs (Appendix C) utilize $\ell_2$ norms while our experiments follow the standard practice of using $l_\infty$ constraints.
>
> **Why we use $\ell_2$ for theory:** $\ell_2$ allows for exact, closed-form derivation of transfer probabilities under random rotations using high-dimensional probability theory (e.g., concentration on the sphere).
>
> **Why the intuition holds for $\ell_\infty$:** The core theoretical insight is **geometric misalignment**. In high dimensions, a random rotation $Q$ maps a direction sensitive to Model A to a direction that is, with high probability, orthogonal to the sensitive directions of Model B. This phenomenon (the 'concentration of measure' in high dimensions) causes random vectors to be orthogonal with high probability regardless of the norm constraint used to define the perturbation ball. The $l_\infty$ constraint defines a hypercube, but if the underlying coordinate bases are rotated relative to each other, the corners of the hypercube (where the optimal $l_\infty$ attack lies) will not align with the sensitive directions of the target model.
>
> # Responding to Weakness 2: VLM Data-Space Attacks (Text) without Images
>
> The reviewer asks why we tested VLM data-space attacks (text jailbreaks) without image inputs. Our goal was to rigorously test the hypothesis that "Data Space transfers, Representation Space does not (unless other considerations are met). To prove the first half of this hypothesis for VLMs, we needed to isolate the Data Space component (text) from the Representation Space component (images).
> -   If we had included images in the text-transfer experiments, a failure to transfer could have been blamed on the image modality interfering (a known phenomenon).
> -   By stripping the image inputs, we demonstrated that the **exact same models** that fail to transfer image attacks (Section 5 results) **successfully transfer text attacks** (Section 5 results).
>
> This isolates the *modality* (Text vs. Vision / Data vs. Representation) as the deciding factor for transferability, rather than the model architecture itself. We will clarify this experimental design choice in the "Methodology" subsection of Section 5.

---

### Official Review · Reviewer_dVF3 · 2025-10-30

**Soundness:** 3
**Presentation:** 3
**Contribution:** 3
**Rating:** 4
**Confidence:** 3

**Summary:**

This paper proposes a fundamental distinction regarding the transferability of machine learning model attacks: attacks in the input (data) space are transferable, while attacks in the model representation space are not, except when the representations across models are geometrically aligned. The study provides both theoretical and empirical insights into this distinction and sheds light on why image jailbreak attacks often fail to transfer between vision-language models (VLMs).

**Strengths:**

1. The overall structure of the paper is clear and coherent, and the content is complete and easy to follow.

2. The research topic is meaningful, and the findings provide valuable explanations for the failure of cross-model transfer in image jailbreak attacks between VLMs.

3. The theoretical analysis is rigorous, with mathematical formulations and proofs that strengthen the paper’s theoretical foundation.

4. The insights presented are thought-provoking and could inspire future work on understanding attack transferability in multimodal and representation-based models.

**Weaknesses:**

1. Figure 1 occupies a large portion of the paper but fails to clearly convey the intended message. It is unclear how the perturbation directions in the second column (data-space perturbations) are chosen. From the figure alone, one cannot intuitively understand the reason why the second column depicts perturbations succeed while the third column shows aligned perturbations that still fail to cross the decision boundary. The decision-space illustration could be significantly improved to more clearly represent these geometric relationships.

2. The paper lacks a formal definition or mathematical formulation of attacks for Image Classifiers, Language Models, and VLMs, especially regarding the distinction between Data-Space Attacks and Representation-Space Attacks. Without this formalization, readers may find it difficult to understand how to construct or evaluate these two types of attacks in practice.

3. For the Image Classifier case, it would be helpful to include a simple derivation for a basic classifier and ensure consistency with the kernel regression analysis framework discussed former.

4. Some experimental figures lack sufficient details about the experimental setup, metrics, and interpretation of the plotted results. Each figure should include a concise explanation of what is being compared, how the axes should be interpreted, and what the main takeaway is.

5. The paper states that “transfer success seemingly does not depend on the number of models used to optimize the attack.” This is confusing, as prior work has shown that ensemble attacks tend to improve transferability. Could the authors provide a theoretical or empirical explanation for this apparent inconsistency?

6. Listing results from different models alone is insufficient to establish the paper’s contribution. The authors should elaborate on the key differences among these models and how these differences substantiate the paper’s theoretical claims. This comparison is essential to understanding the scope and impact of the proposed distinction.

**Questions:**

1. Can the authors elaborate on why attack transferability does not seem to improve with model ensembles, contrary to previous findings? Is it due to representation misalignment, model diversity, or optimization properties?

2. I find this paper presents an interesting and insightful discovery. However, I have several concerns as mentioned above, and I look forward to the authors’ clarifications. If these issues are adequately addressed, I would be willing to raise my score.

---

> ### Author Response · Authors · 2025-11-21
> **Response to Reviewer dVF3**
>
> We thank Reviewer dVF3 for their constructive feedback and for their willingness to raise the score if we address the concerns regarding formalization, figure clarity, and the counter-intuitive results regarding ensembles.
>
> # Responding to Question 1 & Weakness 5: Why Ensembles Do Not Improve Representation-Space Transfer
>
> The reviewer correctly notes that in *data-space* attacks, ensembling usually improves transferability. **Our finding that ensembling fails to improve transfer for representation-space attacks**. This is a key piece of evidence for our geometric misalignment hypothesis.
>
> The Mechanism: Ensembling in data-space works because models trained on similar data often have similar input-output maps that vary slightly and ensembling finds a direction robust to these slight variations.
>
> However, In representation space, the issue is not slight variation in boundaries, but **total basis misalignment**. As shown in our theoretical model (Sec 2), if two models have representations related by a random rotation $Q$, the sensitive directions of Model A are orthogonal to Model B with high probability.
>
> Thus, Optimizing an attack against an ensemble of models ($M_1, M_2, M_3$) in representation space fails to successfully transfer because even if an attack $\delta$ successfully jailbreaks all of the attacked models, the representations of the target model are highly unlikely to be aligned with those of the attacked models. This differs from data space attacks, where the models share similar input-output maps. We will add this explanation to Section 3 and the Discussion.
>
> # Responding to Weakness 2: Formal Definitions of Attacks
>
> We agree that formalizing the distinction for each modality will improve clarity. We will add a "Formal Problem Setup" subsection to Sections 3, 4, and 5.
>
> **General Framework:**
> Let $f(x) = D(E(x))$, where $E$ is an encoder/representation map and $D$ is a decoder/head.
> *   **Data-Space Attack:** Minimize Loss $\mathcal{L}(D(E(x + \delta_{data})), y_{target})$.
> *   **Representation-Space Attack:** Minimize Loss $\mathcal{L}(D(E(x) + \delta_{repr}), y_{target})$.
>
> **Specific Instantiations:**
> *   **Image Classifiers:** $\delta_{data}$ is added to pixels. $\delta_{repr}$ is added to the activation tensor at layer $L$ (e.g., `ResNet.layer3`).
> *   **LLMs:** Data-space attacks (e.g., GCG) optimize discrete tokens $t_{adv}$ appended to $x$. Representation-space attacks optimize a continuous tensor $P$ (soft prompt) added to the embedding sequence: $[P, E(x)]$.
> *   **VLMs:** We treat the image itself as the representation attack. If the LLM is $f_{LLM}$ and the vision encoder is $f_{enc}$, the VLM output is $f_{LLM}(Projector(f_{enc}(Image)))$. We conceptually model the output of the Projector as the perturbation $\delta_{repr}$ entering the LLM's latent space.
>
> # Responding to Weakness 3: Derivation for Image Classifiers
>
> We will explicitly link the classifier case to the kernel regression theory in Section 3.
>
> **Connection:** A deep neural network's penultimate layer features correspond to the mapping $\phi(x)$ in our theoretical model, and the final linear classification layer corresponds to the vector $w$. For a ResNet ending in a linear layer $W$, the logit for class $c$ is $f_c(x) = w_c^T \phi(x)$. A representation attack adds $\delta$ to $\phi(x)$. If a transfer model uses $\tilde{\phi}(x) = Q \phi(x)$ and $\tilde{w}_c = Q w_c$, the transfer is governed by the alignment of $w_c$ and $Q w_c$, exactly matching our Eq. 3 derived for kernel regression.
>
> # Responding to Weakness 1: Improvements to Figure 1
>
> We acknowledge Figure 1 needs work. We will redesign it to clearly show:
>
> 1.  **Data Space:** Two models have different decision boundaries (curved lines), but the *input data points* are in the same coordinate system. A perturbation $\delta$ pushes the point across *both* boundaries because the boundaries rely on the same input features.
>
> 2.  **Representation Space:** We will depict two distinct coordinate systems (e.g., one rotated 90 degrees). A perturbation $\delta$ aligned with the "unstable" direction of Model A points in a "safe" direction for Model B due to the rotation, visually explaining the failure to transfer.
>
> # Responding to Weakness 4 & 6: Figure Details and Model Comparison
>
> **Figure Captions:** We will revise all captions to explicitly state the x/y axes, the metric (ASR or Loss), and the specific models involved, as requested.
>
> **Model Choice:** We selected models to isolate variables:
> -   Same architecture, different seed: Shows effect of random initialization on alignment.
> -   Same base, different finetuning: Shows how SFT preserves alignment (explaining why transfer works here).
> -   Different families: Shows total misalignment.
>
> We will make this selection criteria explicit in the "Experimental Setup" paragraphs.
>
> We believe these revisions will significantly strengthen the paper’s clarity and formal rigor.

---

> > ### Comment · Reviewer_dVF3 · 2025-11-27
> >
> > Thank you for the rebuttal. While I appreciate the clarifications, the core contribution of the paper remains primarily observational, and the proposed distinction feels incremental relative to existing understanding. As such, the improvements do not substantially change my overall assessment, and I will maintain my original score.

---

### Official Review · Reviewer_ERf4 · 2025-10-31

**Soundness:** 1
**Presentation:** 1
**Contribution:** 1
**Rating:** 2
**Confidence:** 5

**Summary:**

This paper attempts to explain the limited transferability of image jailbreaks across Vision-Language Models (VLMs).

**Strengths:**

1.	Investigating and explaining the mechanisms behind image jailbreaks against large models is an important and interesting topic.

**Weaknesses:**

1. The paper fails to provide meaningful insights into the underlying causes of image jailbreak transferability across VLMs.
2. The motivation for analyzing representation-space attacks is unclear. First, the applicability of representation-space attacks is limited. Second, the relationship between representation-space and data-space attacks is not clearly established.
3. Figure 1 does not effectively illustrate the claimed insight of “Why Representation-Space Attacks Fail Where Data-Space Attacks Succeed.”
4. Although the study focuses on VLMs, the authors also analyze other methods such as kernel regression and image classifiers. The rationale for including these analyses and their connection to the main topic is not well explained.

**Questions:**

Please refer to Weaknesses.

---

> ### Author Response · Authors · 2025-11-21
> **Response to Reviewer ERf4**
>
> We thank the reviewer for their candid feedback. We understand that the motivation behind analyzing representation-space attacks and their connection to VLM image jailbreaks was not sufficiently clear in the initial submission. We value the opportunity to clarify this fundamental link, as we believe addressing your concerns effectively highlights the core contribution of our work.
>
> # Addressing the core concern: "Motivation for analyzing representation-space attacks is unclear."
>
> This is the most critical point to clarify. We do not study representation-space attacks as an arbitrary separate category. Rather, our central hypothesis is that visual inputs in adapter-based VLMs effectively function as representation-space attacks, not data-space attacks.
>
> - The Mechanism: In a VLM, the visual input is processed by a vision encoder and a projector before entering the language model. Unlike text (which exists in a shared, discrete "data space" of tokens common across models), the visual features entering the LLM are continuous, high-dimensional, and highly specific to the exact weights of the vision encoder/projector.
>
> - The Insight: Because these internal embeddings are not geometrically aligned across different models (as we show in Sec 6.2 and Fig 17), attacks optimized on one VLM’s visual pipeline fail to transfer to another. This mirrors exactly how "soft prompt" attacks fail to transfer between LLMs (Section 4).
>
> We will revise the Introduction and Section 5 to explicitly state: "We posit that the failure of image jailbreaks to transfer is because the 'data' (images) are treated by the model as unaligned representation injections, unlike text which remains in a shared data space."
>
> # Responding to "Fails to provide meaningful insights into... causes of image jailbreak transferability."
>
> We respectfully disagree, though we acknowledge we must communicate this better. Previous work (Schaeffer et al., 2024; Rando et al., 2024) identified that transfer fails but could not explain why.
>
> We provide the why. Transfer fails because of geometric misalignment in the latent space. We prove this by showing that when we do find models with aligned geometries (via finetuning or specific architectural choices, see Section 6), representation-space attacks do transfer. This confirms that geometric alignment is the control variable for transferability.
>
> # Rationale for including Kernel Regression and Image Classifiers
>
> The reviewer asked why we include other domains. We utilized a scientific "building block" approach to rigorously validate our hypothesis before applying it to the complex VLM setting:
>
>
> 1. Kernel Regression (Theory): We use this to mathematically prove that while functionally identical models always transfer data-space attacks, they rarely transfer representation attacks unless a specific rotation matrix $Q$ aligns them (Theorem A.4). This provides the theoretical foundation.
>
> 2. Image Classifiers and Language Models (Control): We use CNNs and Large Language Models to show this isn't just a VLM phenomenon. Data-space attacks transfer; latent feature attacks (representation-space attacks) do not.
>
> 3. VLMs (Target): We then apply this established theory to solve the mystery of why VLM image attacks don't transfer.
>
> We will make this logical progression explicit at the start of Section 2 to ensure the reader understands these are not unrelated experiments, but necessary steps identifying a fundamental distinction in how attacks are carried out.
>
> #  Improving Figure 1
>
> We agree with the reviewer that Figure 1 can be improved. We will revise it to explicitly label the "Visual Encoder/Projector" path in VLMs as analogous to the "Representation Space" path. We will use color-coding to show that "Data Space" (Text) is shared/aligned between models, while "Representation Space" (Visual Embeddings) is misaligned, visually demonstrating why the attack vector "misses" the target in the second model.
>
> # Conclusion
>
> We hope this clarifies that our analysis of representation space is not a diversion, but the key explanatory mechanism for why VLM attacks fail. We are committed to revising the paper to make this "Data Space = Aligned / Representation Space = Misaligned" distinction the explicit focal point of the narrative. We respectfully ask the reviewer to reconsider the contribution in light of this clarification.

---

> > ### Comment · Reviewer_ERf4 · 2025-11-27
> > **Response to the authors**
> >
> > Thank you for the feedback. After carefully reading the rebuttal, I still have several concerns.
> > First, the practicality of representation-space attacks is limited, as accessing and manipulating internal representations is difficult in real-world applications.
> > Second, the analyses based on kernel regression and image classifiers do not adequately support the claims for VLMs.
> > Finally, the authors have not provided an updated version of Figure 1.
> > Given these remaining issues, I will keep my original score.

---

### Author Response · Authors · 2025-12-03
**Review and Revision Synopsis**

We appreciate the reviewers’ feedback. In this comment we summarise the concerns raised by reviewers and the comprehensive changes we have made to address them. In our revised manuscript, we have incorporated targeted updates—highlighted in blue—that directly address all raised points. Below, we outline each concern alongside the corresponding revisions, including new figures, clearer conceptual explanations, and strengthened methodological formalisation.

The most significant concern, raised by multiple reviewers, was the lack of analysis or justification for why images behave like representation-space attacks for VLMs. Our original draft glossed over the concepts underpinning this argument; we have made substantial revisions to rectify this. We now include a short explanation in the Introduction and in the methodology of Section 5, and most importantly a dedicated subsection in Section 5 that links VLM architecture to the conceptual argument that images function as representation-style inputs. We believe these changes greatly improve the clarity of why adapter-based VLMs cause images to behave as representation-space attacks.  Reviewers ERf4 and dvF3 also expressed concerns about the clarity of Figure 1. We have redesigned the figure to more clearly illustrate the two attack channels (data and representation space) in VLMs, and revised the caption to build intuition for why representation-space attacks do not transfer in the same way. We believe the new figure makes the connection between the two attack types and VLM architecture substantially clearer.  In addition, we have integrated several conceptual and methodological clarifications throughout the manuscript. We now provide a per-section rationale for every experimental setting and ablation we examine (Sections 3–6), explaining why each is necessary to test different aspects of our theoretical claims. We also include a formal problem setup for each experimental setting, as requested by dvF3. Further, we have improved figure captions to clearly describe the experimental setup, metrics, and interpretation of axes (dvF3).

Finally, we have added miscellaneous clarifications throughout the relevant Appendix sections. For the image-classifier experiments, these include an explanation of why ensemble attacks do not succeed in the representation-space setting (dVF3) and a clearer discussion of the gap between the theoretical optimisation constraints and their empirical instantiation (DviV). We also add a short, concrete derivation for a simple classifier that aligns directly with the kernel-regression analysis (D.1.1). For the VLM experiments, we have clarified why we evaluate textual jailbreaks on VLMs even when no textual input is provided (J7K8), emphasising how these attacks operate on the shared representation space within the model’s architecture.
These changes address all reviewer concerns in order to more clearly present our results and insights on the role of latent geometric alignment on attack transferability.

---

### Meta-Review · Area_Chair_oT5h · 2026-01-06

**Summary:**

While the paper addresses an interesting question, the reviewers do not reach consensus on the validity or usefulness of the proposed abstraction. The rebuttal clarifies the authors’ intent but does not sufficiently resolve concerns about conceptual grounding, practical relevance, and evidentiary support. Given the lack of score movement and persistent disagreement, the paper does not meet the acceptance bar in its current form.

In addition, the submission explicitly includes author names and institutional affiliations in the manuscript, which constitutes a clear violation of the double-blind review policy. Regardless of the technical content, this procedural issue alone is sufficient to preclude acceptance. Therefore, I do not recommend accepting this paper.

**Reviewer Concerns:**

The reviews are uniformly negative, with no reviewer providing a positive or clearly supportive score. Reviewers consistently expressed concerns about the paper’s soundness, clarity, and contribution.

**Reviewer Scores:**

Overall, the paper still has substantive issues. While it is possible that one or two reviewers might slightly increase their scores after the rebuttal, this would not change the final recommendation. As stated above, due to the clear violation of the double-blind review policy, I do not recommend accepting this paper.

---

### Decision · Program_Chairs · 2026-01-26

Reject